# Fine-grained List-wise Alignment for Generative Medication Recommendation

**Chenxiao Fan**[1]  **Chongming Gao**[1]*  **Wentao Shi**[1]
**Yaxin Gong**[1]  **Zihao Zhao**[1]  **Fuli Feng**[1]*
[1]University of Science and Technology of China
{simonfan, shiwentao123, gyx2022, zzh1998}@mail.ustc.edu.cn
chongminggao@ustc.edu.cn, fulifeng93@gmail.com

## Abstract

Accurate and safe medication recommendations are critical for effective clinical decision-making, especially in multimorbidity cases. However, existing systems rely on point-wise prediction paradigms that overlook synergistic drug effects and potential adverse drug-drug interactions (DDIs). We propose FLAME, a fine-grained list-wise alignment framework for large language models (LLMs), enabling *drug-by-drug generation* of drug lists. FLAME formulates recommendation as a sequential decision process, where each step adds or removes a single drug. To provide fine-grained learning signals, we devise step-wise Group Relative Policy Optimization (GRPO) with potential-based reward shaping, which explicitly models DDIs and optimizes the contribution of each drug to the overall prescription. Furthermore, FLAME enhances patient modeling by integrating structured clinical knowledge and collaborative information into the representation space of LLMs. Experiments on benchmark datasets demonstrate that FLAME achieves state-of-the-art performance, delivering superior accuracy, controllable safety–accuracy trade-offs, and strong generalization across diverse clinical scenarios. Our code is available at `https://github.com/cxfann/Flame`.

## 1  Introduction

Accurate and safe medication recommendation is essential for clinical decision-making, especially in complex cases involving multimorbidity [1, 2]. In these scenarios, clinicians must consider not only the therapeutic effects of individual drugs, but also their potential interactions and cumulative safety risks. Recent advances in AI have led to the development of automated medication recommendation systems [3, 4, 5], yet their effectiveness remains limited in practice.

Traditional approaches typically adopt a point-wise prediction paradigm, where each drug is evaluated independently based on structured patient data such as diagnoses, procedures, and drug codes [6, 7]. While longitudinal models [8] attempt to capture patient history, they are still constrained by their reliance on discrete labels and lack the capacity to represent unstructured clinical context. More recently, large language models (LLMs) have emerged as a promising solution, due to their strong language understanding and ability to incorporate free-text information such as clinical notes [9, 10]. However, most LLM-based methods still follow the point-wise formulation, constructing the final drug set by aggregating drugs with high predicted scores.

This formulation introduces a fundamental limitation: it overlooks the *synergistic effects and safety constraints* that exist between drugs. In practice, effective prescriptions must balance therapeutic efficacy with the risk of adverse drug-drug interactions (DDIs) [11]. Point-wise models, by design,

---

*Corresponding author

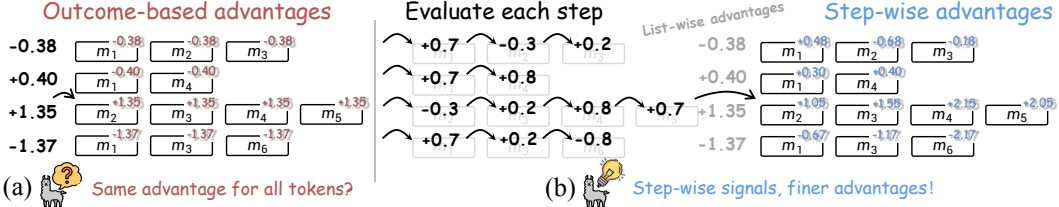

Figure 1: Contrasting advantage computation in GRPO and step-wise GRPO. (a) Outcome-based advantages in GRPO assign a uniform reward to all drugs in a completion. (b) Step-wise GRPO treats the generation as a sequence of medication decisions, where each $m_i$ denotes a distinct drug and simultaneously defines one decision step. Each step is endowed with a potential function, whose change reflects the incremental quality of adding that drug ("Evaluate each step"), and these potential-based signals are aggregated with the list-level outcome reward to yield the final step-wise advantages, enabling finer-grained credit assignment.

cannot account for this interplay. To address this, we argue that medication recommendation should be viewed as a **list-wise decision-making problem**, where the goal is to generate a coherent set of drugs that jointly optimize accuracy and safety.

Existing list-wise methods in NLP often rely on reinforcement learning (RL) techniques to align model outputs with desired properties. One promising approach is Group Relative Policy Optimization (GRPO) [12], which updates model preferences based on relative advantages among a group of candidate outputs. Yet standard GRPO operates at the sequence level (Fig. 1 (a))—assigning a single scalar reward to the entire output—making it difficult to assign credit to individual actions.

To this end, we propose FLAME (**F**ine-grained **L**ist-wise **A**lignment for generative **M**edication r**E**commendation), an LLM-based framework that formulates drug generation as a *drug-by-drug* decision process. At its core is a novel extension of GRPO—**step-wise GRPO**—which models the generation as a sequence of state transitions. Each step adds or removes a drug, and reward shaping is applied via a potential function to provide token-level feedback (Fig. 1 (b)) throughout the generation process. This structure enables FLAME to learn nuanced and controllable decision policies.

To support comprehensive patient modeling, FLAME incorporates multi-source medical knowledge into the LLM via hybrid representations. In addition to the LLM's natural language inputs, we inject structured clinical features (e.g., codes, embeddings from prior models) into the token space, enabling the model to capture both textual context and collaborative signals. We build on Llama3.1-Aloe-Beta-8B [13], a domain-specific LLM with enriched medical knowledge.

We evaluate FLAME on benchmark datasets including MIMIC-III [14], MIMIC-IV [15], and eICU [16]. Experimental results show that FLAME achieves state-of-the-art accuracy while maintaining controllable safety trade-offs. Moreover, FLAME exhibits strong generalization across time and institutions, validating its adaptability to real-world clinical scenarios.

Our contributions are summarized as follows:

- We propose FLAME, an LLM-based list-wise medication recommendation framework that generates prescriptions *drug-by-drug*, explicitly modeling drug interactions and integrating multi-source medical knowledge via hybrid representations.

- We introduce step-wise GRPO, which models medication recommendation as a sequential state transition process and leverages potential-based reward shaping to provide fine-grained feedback for each drug-level decision, enabling controllable and clinically safer recommendations.

- Extensive experiments on benchmark datasets demonstrate that FLAME achieves state-of-the-art performance in accuracy, safety–accuracy trade-offs, and cross-dataset generalization, validating its effectiveness in diverse clinical settings.

## 2   Related Work

In this section, we provide a brief review of medication recommender systems and LLM fine-tuning methods, highlighting their respective strengths and limitations.

## 2.1 Medication Recommendation

Existing medication recommendation methods can be broadly categorized into *instance-based* and *longitudinal* approaches, based on how patient information is modeled.

**Instance-based methods**, such as LEAP [6], treat the medication recommendation task as a multi-instance multi-label learning problem. These approaches typically rely on structured features extracted from a single patient visit. In contrast, **longitudinal methods** incorporate temporal information from patients' hospitalization histories to model long-term disease progression and treatment trajectories. For example, GameNet [7] utilizes hospitalization records and a graph augmented memory module to model DDIs. SafeDrug [3] enhances this by using dual molecular graph encoders to capture drug structural information. COGNet [4] introduces a copy-or-predict mechanism for medication recommendation from historical data, while MoleRec [8] focuses on the relationship between health status and molecular substructures. RAREMed [5] uses a pretrain-finetune framework for structured representation extraction, and NLA-MMR [17] applies a multimodal alignment framework to jointly learn from patient and drug views.

Recently, the development of LLMs has brought new possibilities for medication recommendation. Instead of relying solely on structured codes, these approaches leverage natural language to represent patient conditions more comprehensively. For example, LAMO [10] integrates structured diagnoses and procedures with unstructured textual descriptions of patients, enabling LLMs to model clinical context in a semantically rich manner and generate more personalized drug suggestions.

Many existing methods, including recent LLM-based approaches, still follow the point-wise prediction paradigm [18, 19, 20], assigning independent scores or binary labels to individual drugs while ignoring inter-drug dependencies. In contrast, we formulate medication recommendation as a list-wise decision process and introduce step-wise GRPO for fine-grained reward modeling, integrating medical knowledge, collaborative signals, and LLM-based semantic understanding.

## 2.2 LLM Fine-Tuning Approaches

Fine-tuning LLMs for complex decision tasks has evolved from supervised fine-tuning (SFT) to preference-based reinforcement learning approaches such as reinforcement learning-based fine-tuning (RLHF). While RLHF methods like Proximal Policy Optimization (PPO) [21] offer online adaptability, they incur high overhead due to their multi-component design. Recent advances such as Direct Preference Optimization (DPO) [22] reduce this complexity by leveraging static preference data, but struggle with dynamic optimization.

GRPO [12] further improves training efficiency and policy stability by introducing group-wise relative preference comparisons, achieving strong performance in structured reasoning tasks. However, GRPO remains outcome-based—assigning rewards only to complete outputs—thus failing to capture step-level quality variations within structured decision processes.

Several recent efforts have sought to improve GRPO. DAPO [23] introduces token-level policy-gradient losses to mitigate length bias, ensuring each token contributes equally; however, it still adopts a list-wise advantage formulation, lacking localized reward assignment. StepGRPO [24] decomposes responses into sequential actions and supplements outcome-based evaluation with step-level accuracy and validity rewards. Yet, it continues to aggregate advantages at the list level and often depends on ground-truth process data. While these works enhance signal fidelity and stability, they do not resolve the fine-grained reward allocation problem we address.

On the contrary, step-wise GRPO is a fine-grained alignment method that decomposes generation into decision steps and applies a potential-based reward mechanism to guide each sub-decision. This enables LLMs to better learn the internal logic of complex tasks such as medication recommendation.

## 3 Preliminary

This section formulates the medication recommendation problem through three components: electronic health record, DDI graph, and the optimization goal. We then introduce Group Relative Policy Optimization, a reinforcement learning framework that models outcome-based advantages.

## 3.1 Medication Recommendation

**Electronic Health Records (EHRs).** Each patient's EHR [25] contains both structured and unstructured clinical information and is represented as a sequence of multivariate visits. For a patient $j$ with $V$ historical visits, the EHR is denoted as $\boldsymbol{X}_V^{(j)} = [\mathbf{x}_1^{(j)}, \mathbf{x}_2^{(j)}, \ldots, \mathbf{x}_V^{(j)}]$. Each visit $\mathbf{x}_v^{(j)} \in \boldsymbol{X}_V^{(j)}$ consists of the following components: $\mathbf{x}_v^{(j)} = [(\mathbf{f}_v^{(j)}, \mathbf{n}_v^{(j)}), \mathcal{M}_v^{(j)}]$. where $\mathbf{f}_v^{(j)} \in \{0,1\}^{|\mathcal{F}|}$ is a multi-hot vector encoding structured features from the set $\mathcal{F}$ (e.g., demographics, diagnoses, and procedures), and $\mathbf{n}_v^{(j)}$ is the corresponding unstructured clinical note in natural language. The set $\mathcal{M}_v^{(j)}$ denotes the medications prescribed during the $v$-th visit. For simplicity, we omit the patient index $j$ when it is clear from context.

**Drug-Drug Interaction (DDI) Graph.** The DDI graph [11] models harmful pairwise interactions between medications and is represented as a symmetric binary adjacency matrix $\boldsymbol{D} \in \{0,1\}^{|\mathcal{M}| \times |\mathcal{M}|}$, where $\boldsymbol{D}_{ij} = 1$ indicates a known adverse interaction between medications $m_i$ and $m_j$.

**Task Definition.** Given a patient's historical visit records $\boldsymbol{X}_{V-1}$, as well as the structured fields and unstructured note of the current visit $(\mathbf{f}_V, \mathbf{n}_V)$, the objective is to generate an output $\mathbf{o}_V$, representing the recommended medication set $\mathcal{M}_V$, such that: (1) it closely approximates the ground-truth prescription $\mathcal{M}_{GT}$; and (2) it minimizes the risk of harmful drug-drug interactions as defined by the DDI graph $\boldsymbol{D}$.

## 3.2 Group Relative Policy Optimization (GRPO)

GRPO [12] is a reinforcement learning algorithm that improves policies via group-wise reward normalization. Given a prompt $q \sim P(Q)$ from the task distribution, GRPO samples a group of $G$ candidate outputs $\{\mathbf{o}_i\}_{i=1}^G$ from the current policy $\pi_{\theta_{old}}$. Each output is assigned a scalar reward, and the updated policy $\pi_\theta$ is optimized to favor relatively better responses within the group.

The training objective (omitting clipping for brevity) is:

$$\mathcal{J}_{GRPO}(\theta) = \mathbb{E}_{q, \{\mathbf{o}_i\}} \left[ \frac{1}{G} \sum_{i=1}^G \frac{1}{|\mathbf{o}_i|} \sum_{t=1}^{|\mathbf{o}_i|} \left( \frac{\pi_\theta(\mathbf{o}_{i,t}|\mathbf{o}_{i,<t}, q)}{\pi_{\theta_{old}}(\mathbf{o}_{i,t}|\mathbf{o}_{i,<t}, q)} \hat{A}_{i,t} - \beta D_{KL}[\pi_\theta \| \pi_{ref}] \right) \right], \quad (1)$$

where $\beta$ controls the KL divergence regularization toward a reference policy $\pi_{ref}$.

The token-level advantage $\hat{A}_{i,t}$ is uniformly derived from the normalized reward of the full response:

$$\hat{A}_{i,t} = \tilde{r}_i = \frac{r_i - \text{mean}(\mathbf{r})}{\text{std}(\mathbf{r})}, \quad (2)$$

where $r_i$ is the scalar reward for the $i$-th output, $\mathbf{r} = [r_1, \ldots, r_G]$ is the reward vector for the group.

Although GRPO effectively promotes group-wise preference learning, its outcome-level reward is applied uniformly across all tokens, ignoring variation in token-level quality. This coarse credit assignment hinders fine-grained learning and limits optimization efficiency.

# 4 Method: FLAME

We first introduce step-wise GRPO, the core of FLAME's fine-grained alignment. We then describe the optimization process, followed by the two-stage recommendation framework. Finally, we present the multi-source knowledge fusion strategy.

## 4.1 Step-wise GRPO

**Output Segmentation.** To enable fine-grained credit assignment, we decompose the LLM output into multiple decision steps [26], each corresponding to a semantically coherent token span (e.g., a single medication). Given a generated sequence $\mathbf{o}_i = [o_i^1, o_i^2, \ldots, o_i^{|\mathbf{o}_i|}]$, we segment it into $N_i$ steps:

$$\mathbf{o}_i = \bigcup_{n=1}^{N_i} \mathbf{o}_i^{(n)}, \quad \text{where } \mathbf{o}_i^{(n)} = [o_i^{b_n}, \ldots, o_i^{e_n}]. \quad (3)$$

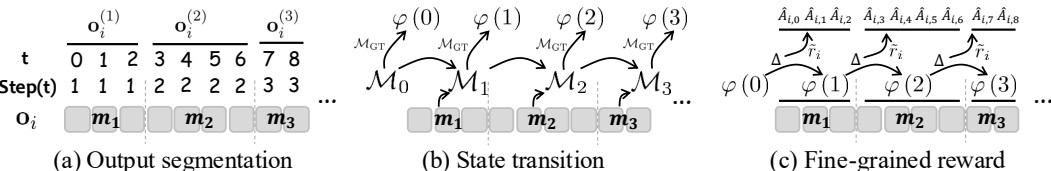

|  | (a) Output segmentation | (b) State transition | (c) Fine-grained reward |

Figure 2: (a) Output is segmented by medication names into decision steps. (b) Each step is viewed as a state, with potentials $\varphi(n)$ derived from comparisons with ground truth $\mathcal{M}_{\text{GT}}$. (c) Potential differences are combined with outcome rewards to provide fine-grained training signals $\hat{A}$.

Each $\mathbf{o}_i^{(n)}$ denotes the token span of the $n$-th step. The process is illustrated in (Fig. 2).

This segmentation allows us to reinterpret generation as a Markov Decision Process, where each step forms a state-action pair. This framing enables localized reward signals that better align step-wise actions with the global output quality.

**Potential-based Reward Shaping.** We extend the standard GRPO advantage (Eq. 2) with a shaping term that captures quality changes between consecutive steps:

$$\hat{A}_{i,t} = \tilde{r}_i + \lambda \cdot F(\text{step}(t), \text{step}(t) - 1) \tag{4}$$

Here, $\text{step}(t)$ maps token $t$ to its step index, and $F(\cdot, \cdot)$ measures step-wise potential difference. $\lambda$ is a weighting coefficient. To preserve policy invariance [27], we define $F$ as the difference of a real-valued potential function $\varphi$:

$$F(n, n - 1) = \gamma \cdot \varphi(n) - \varphi(n - 1) \tag{5}$$

with $\gamma = 1$, assuming equal importance across steps.

**Final Advantage.** The final advantage used for policy optimization becomes:

$$\hat{A}_{i,t} = \tilde{r}_i + \lambda \left( \varphi(\text{step}(t)) - \varphi(\text{step}(t) - 1) \right) \tag{6}$$

This formulation delivers dense training signals aligned with intermediate decision quality [28], encouraging token-level improvements toward globally effective prescriptions.

## 4.2 Fine-grained Alignment for Medication Recommendation

We apply step-wise GRPO to our list-wise medication recommendation task by treating each decision step as a span representing a single medication. We define:

- *State $s_n$*: the patient profile and current medication set $\mathcal{M}_n$.
- *Action $a_n$*: adding or removing a medication $m_n$.

The state transition follows:

$$\mathcal{M}_{n+1} = \begin{cases} \mathcal{M}_n \cup \{m_n\}, & \text{if adding,} \\ \mathcal{M}_n \setminus \{m_n\}, & \text{if removing.} \end{cases} \tag{7}$$

To evaluate the quality of intermediate states, we define a step-level potential function $\varphi(\text{step}(t))$ incorporating: (1) **Correctness**: Jaccard similarity to the ground-truth set $\mathcal{M}_{\text{GT}}$, (2) **Adherence**: constraint violation with respect to the candidate set $\mathcal{M}_C$ (all unique medications in the processed training data), and (3) **Safety**: DDI-based risk from the interaction graph $\boldsymbol{D}$. The potential function is given by:

$$\varphi(\text{step}(t)) = \text{Jaccard}(\mathcal{M}_n, \mathcal{M}_{\text{GT}}) - \alpha \cdot \text{DDI}(\mathcal{M}_n, \boldsymbol{D}) - \beta \cdot \text{RefusalRate}(\mathcal{M}_n, \mathcal{M}_C) \tag{8}$$

where:

$$\text{Jaccard}(\mathcal{M}_n, \mathcal{M}_{\text{GT}}) = \frac{|\mathcal{M}_n \cap \mathcal{M}_{\text{GT}}|}{|\mathcal{M}_n \cup \mathcal{M}_{\text{GT}}|}, \quad \text{RefusalRate}(\mathcal{M}_n, \mathcal{M}_C) = \frac{|\{m \in \mathcal{M}_n \mid m \notin \mathcal{M}_C\}|}{|\mathcal{M}_n|}.$$

$$\text{DDI}(\mathcal{M}_n, \boldsymbol{D}) = \frac{|\{(m_i, m_j) \mid m_i, m_j \in \mathcal{M}_n, i < j, \boldsymbol{D}[m_i, m_j] = 1\}|}{\binom{|\mathcal{M}_n|}{2}}.$$

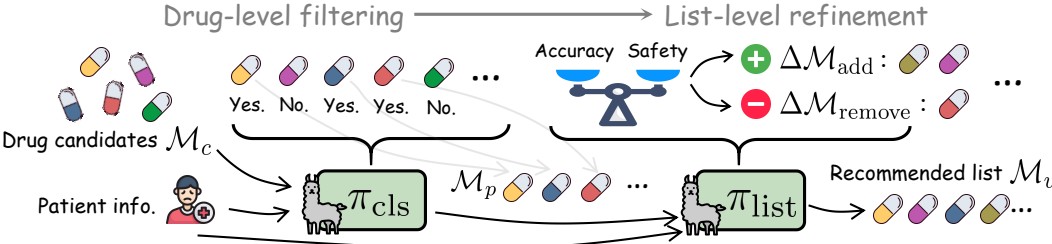

Figure 3: Illustration of the two-stage recommendation framework.

This potential function $\varphi(\text{step}(t))$ allows computing local advantage via step-wise differences (Eq. 6), assigning credit based on incremental improvement in the medication set $\mathcal{M}_n$. The overall list-level reward is defined as the net potential change from the initial to the final step:

$$R_i = \varphi(N_i) - \varphi(0) \tag{9}$$

**Theorem 4.1** (Optimal Policy Equivalence under Reward Reshaping). *Let the token-level reward for each token $t$ in the generated output $\mathbf{o}_i$ be defined under the following two schemes:*

*(1) Outcome-based reward:* $\quad r_{i,t} = \begin{cases} R_i, & \text{if } t \text{ is the terminal token,} \\ 0, & \text{otherwise} \end{cases} \tag{10}$

*(2) Step-wise shaped reward:* $\quad r'_{i,t} = \varphi(\text{step}(t)) - \varphi(\text{step}(t) - 1) \tag{11}$

*Then, both reward formulations yield the same optimal policy.*

The proof is provided in Appendix A.

Using the potential-shaped advantage $\hat{A}_{i,t}$ in Eq. (6), we optimize the policy via step-wise GRPO, while retaining the same objective form as standard GRPO (Eq. (1)).

### 4.3 Two-stage Recommendation Framework

We adopt a cascaded framework to generate an accurate and safe medication set $\mathcal{M}_v$ for a patient's $v$-th visit. The process consists of two stages: drug-level filtering and list-level refinement. The former is implemented via a binary classification model, and the latter through a policy fine-tuned with step-wise GRPO, as illustrated in Fig. 3.

**Drug-level Classifier** ($\pi_{\text{cls}}$). We implement $\pi_{\text{cls}}$ as an LLM-based binary classifier, obtained by SFT the Llama3.1-Aloe-Beta-8B [13] base model, that evaluates the relevance of each candidate drug $m \in \mathcal{M}_c$ given the patient input $x$. The model returns a `Yes`/`No` decision, producing a personalized subset:

$$\mathcal{M}_p = \{m \in \mathcal{M}_c \mid \pi_{\text{cls}}(x, m) = \texttt{Yes}\}. \tag{12}$$

This step establishes personalized relevance by filtering drugs based on individual compatibility prior to joint reasoning.

**List-wise Policy** ($\pi_{\text{list}}$). To support global optimization, we introduce a list-wise policy $\pi_{\text{list}}$ that performs instruction-conditioned edits over $\mathcal{M}_p$. Given patient input $x$, a medication list $\mathcal{M}$, and an instruction (`Add Drug` or `Remove Drug`), the model predicts a modification set:

$$\Delta\mathcal{M} = \pi_{\text{list}}(x, \mathcal{M}, \texttt{Instruction}). \tag{13}$$

We initialize $\pi_{\text{list}}$ with $\pi_{\text{cls}}$, leveraging its drug-patient matching ability, and further adapt it to instruction-driven editing via supervised fine-tuning and step-wise GRPO (see Section 4.1).

**Inference.** At inference, we first apply $\pi_{\text{cls}}$ to obtain $\mathcal{M}_p$ using Eq. (12), then perform list-level edits:

$$\Delta\mathcal{M}_{\text{add}} = \pi_{\text{list}}(x, \mathcal{M}_p, \texttt{Add Drug}), \quad \Delta\mathcal{M}_{\text{remove}} = \pi_{\text{list}}(x, \mathcal{M}_p, \texttt{Remove Drug}). \tag{14}$$

The final recommendation is obtained by applying the edits:

$$\mathcal{M}_v = (\mathcal{M}_p \cup \Delta\mathcal{M}_{\text{add}}) \setminus \Delta\mathcal{M}_{\text{remove}}. \tag{15}$$

This two-stage procedure combines individualized assessment with global reasoning for more controllable and context-aware recommendations. Implementation details, including prompt templates and training setup, are provided in Appendix C.

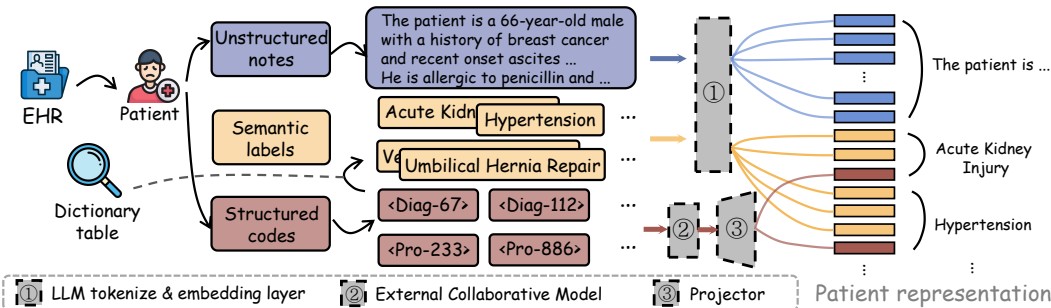

Figure 4: Overview of patient representation construction.

## 4.4 Multi-source Knowledge Fusion

Traditional methods often encode structured inputs (e.g., diagnoses, procedures) into embeddings and integrate external knowledge via architectural modules such as graph encoders [3, 7]. However, LLMs operate primarily on unstructured text, making the incorporation of structured signals non-trivial.

We propose a fusion mechanism that injects collaborative signals from structured sources into LLMs [29, 30], as illustrated in Fig. 4. For each structured entity, we construct two complementary representations: a *textual embedding* $\mathbf{e}^{\text{text}} \in \mathbb{R}^d$ from natural language descriptions, and a *collaborative embedding* $\mathbf{e}^{\text{collab}} \in \mathbb{R}^{d'}$ from a domain-specific encoder. The collaborative embedding is projected into the LLM space via a learnable linear map $\mathcal{P} : \mathbb{R}^{d'} \to \mathbb{R}^d$, yielding:

$$\mathbf{e}^{\text{fused}} = \text{Concat}(\mathbf{e}^{\text{text}}, \mathcal{P}(\mathbf{e}^{\text{collab}})) \tag{16}$$

This fusion enriches LLM inputs with high-level domain signals. We use four types of collaborative embeddings: **(1) Patient-level:** from RAREMed [5], a pretrained model encoding clinical context via a transformer. **(2) Diagnosis-level** and **(3) Procedure-level:** from MICRON [31], capturing co-occurrence patterns from EHRs. **(4) Medication-level:** from Mole-BERT [32], modeling drug substructure–function relationships via molecular graphs.

Moreover, we adopt **Llama3.1-Aloe-Beta-8B** [13], a medical LLM, as our backbone, enabling the fusion of structured embeddings with textual semantics. This hybrid representation not only enhances the model to capture fine-grained clinical signals but also benefits from the comprehensive medical knowledge encoded in the LLM, together forming our multi-source knowledge fusion framework.

## 5 Experiments

We begin by detailing the experimental setup. We then evaluate FLAME against baselines across three dimensions: accuracy, safety–accuracy trade-offs, and cross-dataset generalization. Finally, an ablation study assesses the contribution of each component in our framework.

### 5.1 Experimental Settings

**Datasets.** We use real-world EHR datasets: MIMIC-III [14] for training and evaluation, and MIMIC-IV [15] and eICU [16] for generalization testing. DDI relations are obtained from TWOSIDES [11]. Following prior works, MIMIC-III is split into training/validation/test sets (4:1:1).

**Implementation.** All methods are implemented in PyTorch and trained on NVIDIA A100 GPUs. Details on preprocessing, hyperparameters, and prompts are in Appendices B and C.

**Baselines.** We compare FLAME with: (1) Traditional methods: LEAP [6], GAMENet [7], Safe-Drug [3], COGNet [4], MICRON [31], MoleRec [8], NLA-MMR [17], RAREMed [5]; (2) LLM-based method: LAMO [10].

**Metrics.** Correctness is evaluated by Jaccard similarity and F1 score; safety by DDI rate. Definitions and calculation details are in Appendix C.

## 5.2 Overall Performance Comparison

We evaluate FLAME from three perspectives: **correctness**, **safety controllability**, and **generalizability**. EHR datasets contain inherent DDI risks (e.g., 13.69% in MIMIC-III), meaning data fitting does not guarantee safe prescriptions. Minimizing DDIs often compromises accuracy, making the correctness–safety trade-off a key metric for practical utility.

Unlike prior works reporting a single performance point, we explicitly decouple correctness and safety in evaluation, enabling a clearer assessment of a model's reasoning ability and controllability under varying safety constraints. To assess generalization, we further test on MIMIC-IV and eICU, examining performance under distribution shifts beyond the training domain.

**Correctness Comparison**. Table 1 reports correctness results on MIMIC-III. Instance-based models (e.g., LEAP) perform poorly due to a lack of historical context modeling. Longitudinal models like SafeDrug and GAMENet improve performance by leveraging temporal data, while MoleRec further enhances accuracy by incorporating molecular substructure information. RAREMed achieves stronger results with transformer-based clinical representations. The LLM-based LAMO benefits from unstructured clinical notes but lacks structured co-occurrence knowledge, limiting its potential. Our proposed FLAME outperforms all baselines in Jaccard and F1 scores by formulating medication recommendation as a list-wise decision problem and fusing multi-source clinical knowledge.

**Safety Control Comparison**. We evaluate controllable safety by adjusting the penalty parameter $\alpha$ in Eq. 8, generating recommendations under varying DDI constraints. Fig. 5(a) shows Jaccard–DDI trade-off curves. Among models supporting safety control, FLAME consistently achieves better accuracy–safety balance than baselines. Notably, methods like LAMO are excluded as they lack safety adjustment mechanisms. These results validate the effectiveness of step-wise GRPO in enabling controllable, clinically safer recommendations.

**Generalization Comparison**. We assess generalization through temporal and external validations. For temporal validation, models trained on MIMIC-III are evaluated on MIMIC-IV, which covers newer time periods and ICD-10 codes. Fig. 5(b) shows that LLM-based models (FLAME, LAMO) degrade less over time compared to structured-data baselines, highlighting the robustness of language-based representations to coding shifts. FLAME further surpasses LAMO due to its integration of multi-source knowledge and step-wise reasoning.

Table 1: Correctness comparison on MIMIC-III. #Med indicates the average number of prescribed medications per patient (ground truth: 23.43).

| Model | Jaccard | F1 | #Med |
|---|---|---|---|
| LEAP | 0.3073 | 0.4610 | 19.69 |
| SafeDrug | 0.3556 | 0.5135 | 22.94 |
| GAMENet | 0.3851 | 0.5422 | 20.02 |
| COGNet | 0.3876 | 0.5458 | 28.45 |
| MICRON | 0.3843 | 0.5419 | 20.49 |
| NLA-MMR | 0.3867 | 0.5453 | 25.54 |
| MoleRec | 0.4081 | 0.5677 | 24.73 |
| RAREMed | 0.4174 | 0.5776 | 23.03 |
| LAMO | 0.4701 | 0.6294 | 23.66 |
| **FLAME** | **0.4836** | **0.6408** | 22.24 |

For external validation, we evaluate on eICU, a multi-center dataset with distributional and coding differences. As shown in Fig. 5(c), LLM-based methods again outperform structured-input baselines. FLAME achieves superior out-of-distribution performance, demonstrating the advantage of combining list-wise decision modeling with collaborative knowledge fusion.

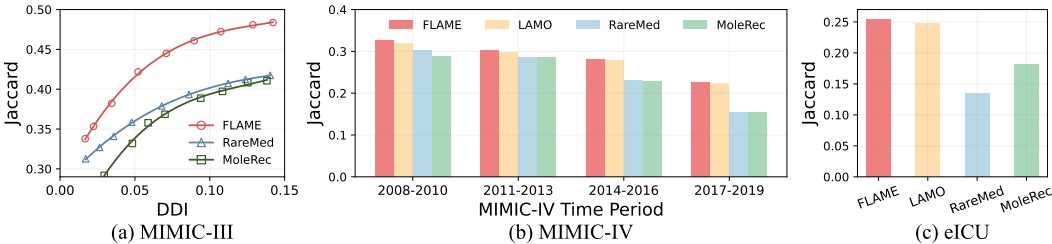

Figure 5: Comparison with strong baselines. (a) Safety–accuracy trade-off on MIMIC-III. (b) Temporal generalization from MIMIC-III to MIMIC-IV. (c) External generalization to eICU.

Table 2: Ablation study. "w/o" denotes "without".

| Model Variants | Jaccard | F1 |
|---|---|---|
| (a) w/o $\pi_{\text{cls}}$ | 0.3473 | 0.5024 |
| (b) w/o $\pi_{\text{list}}$ | 0.4785 | 0.6354 |
| (c) w/o SFT in $\pi_{\text{list}}$ | 0.4810 | 0.6382 |
| (d) w/o step-wise GRPO in $\pi_{\text{list}}$ | 0.4789 | 0.6367 |
| (e) with standard GRPO | 0.4813 | 0.6385 |
| (f) w/o structured codes | 0.4535 | 0.6122 |
| (g) w/o unstructured notes | 0.4104 | 0.5692 |
| (h) w/o $e^{\text{text}}$ | 0.3930 | 0.5509 |
| (i) w/o $\tilde{e}^{\text{collab}}$ | 0.4448 | 0.6080 |
| (j) with $\tilde{e}_{\text{rand}}^{\text{collab}}$ | 0.4653 | 0.6231 |
| (k) FLAME-LLaMA2 | 0.4591 | 0.6168 |
| (l) FLAME-LLaMA3 | 0.4774 | 0.6344 |
| **FLAME** | **0.4836** | **0.6408** |

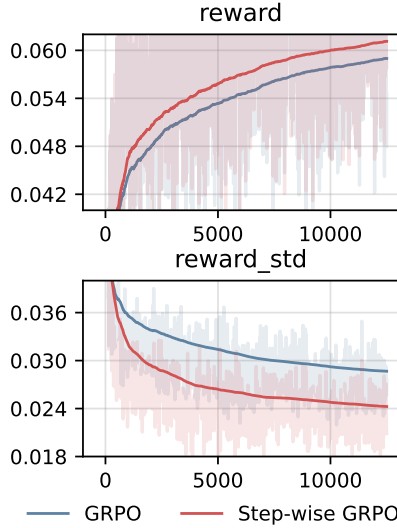

Figure 6: Training curves of GRPO and step-wise GRPO.

## 5.3 Ablation Study

We conduct ablations to assess the impact of each component in FLAME, including the list-wise decision model, step-wise GRPO, and multi-source knowledge fusion.

**List-wise Decision Model.** We first examine the roles of $\pi_{\text{cls}}$ and $\pi_{\text{list}}$. Table 2 (a)-(b) shows that removing $\pi_{\text{cls}}$ significantly harms performance, highlighting the importance of individualized filtering. Excluding $\pi_{\text{list}}$ also degrades results, indicating that list-wise modeling is essential for capturing decision structures beyond point-wise analysis.

We further evaluate the two-stage training of $\pi_{\text{list}}$. Skipping SFT limits task-form alignment, while omitting step-wise GRPO weakens preference learning (Table 2 (c)-(d)). These results confirm the necessity of both SFT and step-wise preference alignment.

**Step-wise GRPO.** Replacing step-wise GRPO with standard GRPO (Table 2 (e)) results in clear performance drops, demonstrating that outcome-only rewards are insufficient. Fig. 6 shows that step-wise GRPO provides denser feedback, accelerating learning and improving stability.

**Multi-source Knowledge Fusion.** Integrating structured codes and unstructured clinical notes significantly boosts performance; removing either degrades results (Table 2 (f)-(g)). To analyze embedding strategies, we replace collaborative embeddings with random vectors $\tilde{e}_{\text{rand}}^{\text{collab}}$. While random embeddings offer slight gains over text-only inputs, they lack meaningful domain knowledge, resulting in a notable gap from FLAME (Table 2 (h)-(j)).

**LLM Backbone.** Replacing the Llama3.1-Aloe-Beta-8B backbone with general large language models (LLaMA2 and LLaMA3) leads to noticeable performance drops (Table 2 (k)-(l)). This highlights the critical role of domain-specific medical knowledge embedded in Llama3.1-Aloe-Beta-8B for accurate medication recommendation.

## 5.4 In-Depth Analysis

To further understand the empirical behaviors of FLAME, we conduct in-depth analyses on the necessity of the list-wise refinement policy $\pi_{\text{list}}$, and the effectiveness of step-wise GRPO. These analyses complement the quantitative results and provide qualitative evidence for our design choices.

**Role of the List-wise Policy $\pi_{\text{list}}$.** In Table 2(b), removing $\pi_{\text{list}}$ results in a relatively small numerical drop (Jaccard 0.4836 → 0.4785), which may appear marginal at first glance. However, this is expected, since $\pi_{\text{list}}$ acts as a *lightweight refinement module* rather than a primary predictor. On the MIMIC-III test set, $\pi_{\text{cls}}$ outputs lists of average length 20.83, while $\pi_{\text{list}}$ performs only ∼2.03

add/remove edits per patient on average. These small-scale edits are crucial: they primarily resolve subtle drug–drug dependencies and redundancies that point-wise classifiers cannot capture.

We identify two major benefits of $\pi_{\text{list}}$. (1) *Safety–accuracy controllability.* $\pi_{\text{cls}}$ is optimized to mimic ground truth prescriptions, producing DDI rates (0.1336) close to the real data distribution (0.1369). While effective for data fitting, this limits its ability to adapt to stricter safety constraints required in real-world clinical practice. In contrast, $\pi_{\text{list}}$ incorporates the safety penalty $\alpha$ (Eq. 8), enabling explicit and controllable adjustment of safety levels. (2) *Modeling drug dependencies.* Since $\pi_{\text{cls}}$ predicts each drug independently, it cannot capture co-prescription patterns or redundancies. For instance, *Piperacillin* (ID 130) is a broad-spectrum $\beta$-lactam antibiotic overlapping in indication with *Cefepime* (ID 92), *Meropenem* (ID 45), and *Ampicillin* (ID 113). In such cases, $\pi_{\text{cls}}$ may recommend redundant combinations or omit suitable alternatives. By performing list-level reasoning, $\pi_{\text{list}}$ corrects these issues effectively: across the test set, it edited 28 Piperacillin-containing lists with an 89% correctness rate, demonstrating its capacity for meaningful refinement beyond point-wise learning.

Table 3: Examples of $\pi_{\text{list}}$ refinements. IDs **45**, **92**, **113**, and **130** denote antibiotics with overlapping effects. Correct, over-predicted, and missing drugs are colored green, red and blue, respectively. Drugs in parentheses indicate those absent from $\mathcal{M}_p$ but present in the ground truth list.

| Case ID | $\pi_{\text{cls}}$ Output $\mathcal{M}_p$ | $\pi_{\text{list}}$ Refinement |
|---|---|---|
| 266 | 30, 76, **113**, 137, 6, 74, **130**, (14, 104, 114), ... | add: 14, 114; remove: **130** |
| 887 | 2, 71, 99, 8, 22, **130**, (34, 48, 57, 80, **92**, 126), ... | add: 34, 48, 57, **92**; remove: **130** |

As shown in Table 3, $\pi_{\text{list}}$ rectifies redundant and missing predictions by exploiting list-wise signals. In Case 266, $\pi_{\text{cls}}$ simultaneously predicts IDs 113 and 130, which share overlapping effects, while omitting several relevant items. $\pi_{\text{list}}$ removes the redundant 130 and adds the missing entries, thereby enhancing the precision of the generated medication list. In Case 887, the point-wise model again over-predicts 130 and misses 92, two drugs with overlapping indications, while $\pi_{\text{list}}$ corrects both errors through list-level reasoning. These examples demonstrate that even lightweight list-wise edits can yield clinically coherent refinements by leveraging inter-drug dependencies absent in point-wise prediction, confirming the essential role of $\pi_{\text{list}}$ beyond its modest aggregate numerical gain.

**Effectiveness of Step-wise GRPO.** Step-wise GRPO is the optimization backbone for $\pi_{\text{list}}$. While the absolute improvement over standard GRPO in Table 2(e) is modest, the relative effect is substantial: list-wise refinement's advantage over point-wise classification increases from 0.0028 (standard GRPO) to 0.0051 (step-wise GRPO), an 82.1% relative gain. Additionally, Fig. 6 shows faster reward convergence and more stable training dynamics under step-wise GRPO, owing to its denser intermediate feedback rather than reliance on terminal rewards—an important trait for long edit sequences such as medication list refinement.

# 6   Conclusion

We presented FLAME, an LLM-based list-wise medication recommendation framework that formulates prescription generation as a sequential *drug-by-drug* decision process. By introducing **step-wise GRPO** with potential-based reward shaping, FLAME enables fine-grained credit assignment, effectively capturing drug interactions and enforcing safety constraints such as DDIs. Through hybrid representations that fuse structured clinical data, collaborative signals, and unstructured textual information, FLAME enhances the LLM's capacity for accurate and clinically grounded recommendations. Extensive experiments demonstrate FLAME's superior accuracy, controllable safety–accuracy trade-offs, and strong generalization across temporal and institutional shifts. These results highlight the potential of fine-grained list-wise alignment for building reliable, adaptable clinical decision support systems. In future work, we plan to further improve FLAME's generalization in diverse and evolving healthcare environments, advancing towards scalable and equitable AI-driven solutions for real-world clinical practice. In addition, we will explore the integration of physician decisions with FLAME to build a human-in-the-loop medication recommendation system.

## Acknowledgments

This work is supported by the National Natural Science Foundation of China (62272437,62402470), the University Synergy Innovation Program of Anhui Province (GXXT-2023-071), Anhui Provincial Natural Science Foundation (2408085QF189), and the advanced computing resources provided by the Supercomputing Center of the USTC.

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

# A  Theoretical Analysis

*Proof.* We prove the theorem using the reward shaping framework introduced by [27], which shows that augmenting a reward function with a potential-based shaping term does not alter the optimal policy.

**Lemma A.1** (Ng et al., 1999). *Let $r(s_t, a_t)$ and $r'(s_t, a_t)$ be two reward functions. If there exists a potential function $\Phi(s)$ such that:*

$$r'(s_t, a_t) = r(s_t, a_t) + \gamma \Phi(s_{t+1}) - \Phi(s_t), \tag{17}$$

*where $\gamma$ is the discount factor, then the optimal policies under $r$ and $r'$ are identical.*

Now, consider the outcome-based reward:

$$r_{i,t}^{\text{terminal}} = \begin{cases} \varphi(J_i), & \text{if } t \text{ is terminal}, \\ 0, & \text{otherwise}. \end{cases}$$

We define the shaped step-wise reward as:

$$r_{i,t}^{\text{step}} = \varphi(\text{step}(t)) - \varphi(\text{step}(t) - 1).$$

Let us define a potential function $\Phi(t) = \varphi(\text{step}(t))$. Then the step-wise reward can be rewritten as:

$$r_{i,t}^{\text{step}} = \Phi(t) - \Phi(t - 1).$$

Summing over the episode:

$$\sum_{t=1}^{T} r_{i,t}^{\text{step}} = \Phi(T) - \Phi(0) = \varphi(J_i) - \varphi(0),$$

assuming $\text{step}(T) = J_i$. Since $\varphi(0)$ is constant across all trajectories, it does not affect the optimization. Thus, both reward schemes only differ by a constant shift and a potential-based shaping term.

By the lemma, such a transformation does not change the optimal policy.

**Conclusion:** The outcome-based terminal reward and the shaped step-wise reward are equivalent in terms of the policies they induce.  □

# B  Dataset Details

## B.1  Data Preprocessing

We follow the data preprocessing procedures adopted in prior work [10], using structured and unstructured EHR data from the MIMIC-III [14] database. For structured clinical information, we extract diagnosis and procedure codes and retrieve their corresponding textual descriptions from the ICD dictionary tables. Since the original long titles in the dictionary can be overly verbose and difficult for LLMs to process, we utilize GPT-4o to compress them into concise titles while preserving the essential semantic content. This compression strategy follows the same approach as adopted in LAMO [10] to ensure semantic integrity and model compatibility.

For medication data, we map National Drug Codes to DrugBank IDs and obtain associated text descriptions from corresponding lookup tables. To incorporate unstructured information, we follow the previous pipeline and extract segments from clinical notes, including **History of Present Illness**, **Past Medical History**, **Allergies**, and **Medications on Admission**, using GPT-4o as a parser to construct patient condition descriptions. A representative example of unstructured text extracted from clinical notes is shown below:

> **Example of unstructured text extracted from clinical notes**
>
> **History of Present Illness:** The patient is a 66-year-old female with a history of breast cancer and recent onset ascites and pelvic mass. She presented with worsening abdominal discomfort, poor oral intake, and dyspnea.
>
> **Past Medical History:** The patient has a history of breast cancer, hypertension, hypothyroidism, tubal ligation, and a previous metacarpal fracture.
>
> **Allergies:** [codeine].
>
> **Medications on Admission:** [Lisinopril, Effexor, Levoxyl, Tamoxifen, Fosamax].

## B.2 Dataset Statistics

After preprocessing the structured clinical codes (diagnoses, procedures, and medications), we obtain the dataset statistics summarized in Table 4.

Table 4: Statistics of processed data in MIMIC-III

| Items | MIMIC-III |
|---|---|
| # of visits / # of patients | 14207 / 6226 |
| dis. / prod. / med. space size | 1676 / 511 / 151 |
| avg. / max # of visits | 2.28 / 29 |
| avg. / max # of dis. per visit | 13.59 / 39 |
| avg. / max # of pro. per visit | 4.23 / 27 |
| avg. / max # of med. per visit | 23.36 / 77 |

## C  Experimental Setup

### C.1  Training Details and Hyperparameters

We conduct all experiments on NVIDIA A100-SXM4-80GB GPUs, with Python 3.10 and PyTorch 2.5.1. During all training stages, the model is quantized using bf16 and LoRA, and optimized using the `adamw_torch` optimizer.

For the SFT of drug-level classifier $\pi_{cls}$, we use Llama3-Aloe-8B-Alpha as the base model. Four projectors are randomly initialized: pat_projector, diag_projector, pro_projector, and med_projector, each consisting of a two-layer MLP with GELU activation. The learning rate is set to 5e-4, with a batch size of 128 and one epoch.

For the SFT of list-wise policy $\pi_{list}$, the model and projector weights from the previous step ($\pi_{cls}$) are used as initialization. The learning rate remains 5e-4, with a batch size of 64 and one epoch.

When performing step-wise GRPO on $\pi_{list}$, we initialize the model and projector weights from the previous SFT step. The projector parameter 'r_grad' is set to False. We use the Unsloth framework and vLLM 0.7.3 for acceleration. The learning rate is set to 1e-5, with a batch size of 16, 'num_generations' set to 8, and one epoch. The hyperparameter $\alpha$ is chosen from the set [0, 2, 5, 10, 20, 30, 40, 50] to adapt to different DDI requirements, while $\beta$ is set to 0.5, and $\lambda$ is set to 5.

### C.2  Baseline Methods and Evaluation Metrics

We provide a comprehensive overview of baseline models that have been widely used for the medication recommendation task:

- **LEAP** [6] formulates the medication prediction task using only the current visit and models label dependencies via multi-instance multi-label learning.

- **GAMENet** [7] integrates EHR and DDI graphs via graph-augmented memory networks to ensure both accuracy and safety.

- **SafeDrug** [3] employs dual molecular encoders to learn molecular representations and improve safe prescription decisions.
- **COGNet** [4] leverages a copy-or-predict mechanism to incorporate historical prescriptions into current decision making.
- **MICRON** [31] introduces a recurrent residual learning model that treats medication change dynamics as the primary learning target.
- **MoleRec** [8] proposes a substructure-aware attention mechanism that models drug substructure–patient condition interactions at a fine-grained level.
- **RAREMed** [5] utilizes a Transformer-based encoder to derive comprehensive patient representations and mitigate fairness issues in recommendation.
- **NLA-MMR** [17] designs a multi-modal alignment framework to jointly encode patient and medication representations across modalities.
- **LAMO** [10] builds on LLaMA-2 and incorporates unstructured clinical notes to better capture the patient's condition, leveraging the semantic understanding capabilities of LLMs for accurate medication recommendation.

We adopt Jaccard Score and F1 Score to evaluate the prediction accuracy, and DDI rate to assess the safety of recommended medication sets. Given the model prediction $\mathcal{M}_v$ and the ground-truth set $\mathcal{M}_{\mathrm{GT}}$, we define:

- **Jaccard Score**:

$$\mathrm{Jaccard}(\mathcal{M}_v, \mathcal{M}_{\mathrm{GT}}) = \frac{|\mathcal{M}_v \cap \mathcal{M}_{\mathrm{GT}}|}{|\mathcal{M}_v \cup \mathcal{M}_{\mathrm{GT}}|} \tag{18}$$

- **F1 Score**, computed based on precision and recall:

$$\mathrm{Precision} = \frac{|\mathcal{M}_v \cap \mathcal{M}_{\mathrm{GT}}|}{|\mathcal{M}_v|}, \quad \mathrm{Recall} = \frac{|\mathcal{M}_v \cap \mathcal{M}_{\mathrm{GT}}|}{|\mathcal{M}_{\mathrm{GT}}|}, \quad \mathrm{F1} = \frac{2 \cdot \mathrm{Precision} \cdot \mathrm{Recall}}{\mathrm{Precision} + \mathrm{Recall}} \tag{19}$$

- **DDI rate** measures the ratio of harmful drug-drug interactions among predicted medications:

$$\mathrm{DDI}(\mathcal{M}_v, \boldsymbol{D}) = \frac{|\{(d_i, d_j) \mid d_i, d_j \in \mathcal{M}_v, \, i < j, \, \boldsymbol{D}[d_i, d_j] = 1\}|}{\binom{|\mathcal{M}_v|}{2}} \tag{20}$$

### C.3 Prompt Templates

In Section 4.3, we propose a two-stage recommendation framework, which consists of: (1) drug-level filtering and (2) list-level refinement. Below we provide the prompt templates used in each stage.

For the drug-level classifier ($\pi_{\mathrm{cls}}$), we formulate a binary classification task over each candidate drug $m \in \mathcal{M}_c$ in the set of drug candidates. Each drug is evaluated independently based on its appropriateness for the given patient. The model is prompted with the patient's clinical representation and a textual description of the candidate drug. Specifically, {{Patient Representation}} includes both patient's current status and previous hospitalization records. {{Drug Candidate}} refers to the textual representation of a single drug under evaluation.

---

**Prompt Template for Drug-level Filtering**

You are about to evaluate a candidate drug for a patient's clinical condition. You will be provided with the patient's current condition, as well as information about their previous hospitalization, and the candidate drug. Your task is to determine whether the candidate drug is effective and safe for the patient. Please respond with <Yes.> or <No.> without providing an explanation.
{{Patient Representation}}
Candidate drug: {{Drug Candidates}}
###Answer:

---

For the list-level refinement policy ($\pi_{\text{list}}$), we further optimize the drug set $\mathcal{M}_p$ produced by the drug-level classifier by modeling dependencies among drugs and refining the list through two complementary tasks: (1) adding missing yet clinically necessary drugs, and (2) removing drugs that are unnecessary or potentially unsafe for the patient. Each task is formulated as a separate prompt, where the model is asked to revise a pre-predicted drug list based on the patient's condition and medical history.

The task instruction is injected into the prompt through the `{{instruction}}` placeholder, with the following two variants:

- **Add task:** add any relevant drugs that are missing from the pre-predicted list.
- **Remove task:** remove any drugs from the pre-predicted list that are unnecessary or unsuitable for the patient's condition.

The unified prompt template used in both tasks is shown below:

---

**Prompt Template for List-level Refinement**

You are tasked with refining a drug list for a patient's clinical condition. You will be provided with the patient's condition, information about their previous hospitalization and a pre-predicted drug list.
Your task is to: `{{instruction}}`
Please output a list of drugs that need to be `{{added or removed}}`, with each drug separated by a newline.
`{{Patient Representation}}`
###Pre-predicted Drug List: `{{`$\mathcal{M}_p$`}}`
###Drugs to `{{Add or Remove}}`:

---

To enable comprehensive modeling of the patient's condition within each task prompt, we incorporate both current and historical clinical information into the `{{Patient Representation}}` field whenever available. Specifically, we differentiate between patients with and without prior hospitalization records. The formatting is as follows:

---

**Prompt Template for Patient Representation**

**For patients with previous hospitalization records:**
### Previous Hospitalization: `{{Previous Representation}}`
### Current Clinical Condition: `{{Current Representation}}`

---

**For patients without previous hospitalization records:**
### Current Clinical Condition: `{{Current Representation}}`

---

For the patient's previous hospitalization representation `{{Previous Representation}}`, we construct a hybrid representation that integrates structured information from diagnoses, procedures, and medications, along with demographic data.

---

**Prompt Template for Previous Representation**

Age: `{{Age of the patient, eg. 68}}`,
Gender: `{{Patient's biological sex, e.g., Male}}`,
Diagnoses: `{{Hybrid diagnosis representation}}`,
Procedures: `{{Hybrid procedures representation}}`,
Drug names: `{{Hybrid drugs representation}}`.

---

Here we further describe the format of the `{{Current Representation}}`, which captures the patient's clinical status at admission. This representation integrates both structured data—such as diagnoses and procedures—and unstructured information extracted from clinical notes, including the history of present illness, past medical history, allergies, and medications on admission.

> **Prompt Template for Current Representation**
>
> Patient representation: `{{Latent embedding}}`,
> History of present illness: `{{Unstructured text extracted from clinical notes}}`,
> Past medical history: `{{Unstructured text extracted from clinical notes}}`,
> Allergies: `{{Unstructured text extracted from clinical notes}}`,
> Medications on admission: `{{Unstructured text extracted from clinical notes}}`,
> Diagnoses: `{{Hybrid diagnosis representation}}`,
> Procedures: `{{Hybrid procedures representation}}`.

As introduced in Section 4.4, our model leverages hybrid representations to encode structured EHR fields—namely diagnoses, procedures, and medications—by combining the textual representations generated by LLM with collaborative embeddings produced by domain-specific encoders. These hybrid representations correspond to the placeholder fields—`Hybrid diagnosis representation`, `Hybrid procedures representation`, and `Hybrid drugs representation`—previously introduced in our prompt templates. Here, we provide concrete examples of how each field is instantiated.

Specifically, for each structured medical code, we first obtain a textual description suitable for LLM input. In parallel, we apply a domain-specific encoder to the original structured code to extract a collaborative embedding, which captures latent clinical semantics. This collaborative embedding is then projected into the LLM embedding space using a learnable adapter, rather than being passed through the LLM's standard tokenizer and embedding layer. The final hybrid representation is thus composed of both the textual token sequence and its aligned embedding enhancement. More details of this embedding alignment can be found in Section 4.4.

The following illustrates the prompt-level instantiation of hybrid representations:

> **Prompt-level Instantiation of Hybrid Representations**
>
> Diagnoses: Hematochezia **[DiagEmb-12]**, Acute Kidney Injury **[DiagEmb-56]**, Heart Failure **[DiagEmb-104]** ...
> Procedures: Coronary Artery Stenting **[ProEmb-8]**, Three Vessel Procedure **[ProEmb-79]**, Stent Insertion **[ProEmb-226]** ...
> Drug names: Calcium gluconate **[DrugEmb-2]**, Captopril **[DrugEmb-31]**, Magnesium sulfate **[DrugEmb-3]** ...

# D   Limitations

While FLAME shows promising results, it also has several limitations that warrant further investigation:

- **Computational cost and deployment challenges:** The use of LLM introduces significant training and inference overhead. This can limit the practical deployment of the system in resource-constrained clinical settings, especially in real-time or on-device applications.

- **Lack of automated multi-modal data integration:** Although our framework incorporates both structured and unstructured patient data, it currently lacks an end-to-end automated mechanism for multi-modal data processing. In particular, the handling of unstructured patient narratives relies on manually invoking external APIs (e.g., GPT-4o), which limits scalability and consistency.

- **Static evaluation setting:** Our model is trained and evaluated on static EHR datasets, which do not fully capture the dynamic nature of real-world clinical workflows. Bridging this gap requires further investigation into how such models can be adapted for continual learning or integration with live clinical decision support systems.

# E   Broader Impacts

This work presents the potential for significant positive societal impact by advancing the integration of LLMs into clinical decision-making. By providing accurate, context-aware, and safe medication recommendations, our framework supports physicians rather than replacing them, offering interpretable suggestions that can complement and enhance clinical judgment. This human-in-the-loop design reinforces trust and accountability in medical AI systems, ensuring that final decisions remain under professional oversight. Furthermore, the model's ability to handle both structured clinical data and unstructured free-text inputs aligns naturally with real-world clinical workflows, where critical patient information often resides in narrative form. This flexibility reduces the burden of data curation and enables more seamless adoption in practice. In addition, the model's demonstrated generalizability across time and institutional boundaries opens avenues for deployment in under-resourced settings, where access to experienced clinicians and high-quality clinical decision support is limited. By bridging this gap, our work contributes toward improving the equity and reach of healthcare services. Ultimately, we envision this research as a step toward scalable, adaptive, and collaborative AI systems that respect clinical complexity while expanding access to safe and effective medical support across diverse populations.

However, despite its potential benefits, the deployment of FLAME also poses certain societal risks. First, the model's performance may be affected by biases in the training data, which could lead to disparities in recommendation quality across different patient populations, particularly those underrepresented in historical records. Second, there is a risk of misuse if such systems are deployed without proper clinical oversight—for example, being used in unauthorized settings or for non-medical purposes, potentially leading to harmful or misleading recommendations. These concerns underscore the need for responsible deployment practices, transparency, and safeguards to ensure that FLAME is used ethically and equitably in real-world clinical environments.

