# OpenReview forum: "Fine-grained List-wise Alignment for Generative Medication Recommendation"
_NeurIPS.cc/2025/Conference — NeurIPS 2025 spotlight_

### Official Review · Reviewer_2sXp · 2025-06-18

**Clarity:** 3
**Significance:** 3
**Originality:** 3
**Rating:** 5
**Confidence:** 4

**Summary:**

The paper introduces **FLAME**, a medication recommendation framework that treats prescription generation as a **step-by-step list-wise decision process** using large language models. It proposes **step-wise Group Relative Policy Optimization (GRPO)** to assign fine-grained rewards for each drug added or removed, improving both accuracy and safety. FLAME also fuses structured and unstructured clinical data to enhance patient modeling. Experiments on multiple EHR datasets show that FLAME outperforms prior methods in multiple metrics.

**Questions:**

1. **Clinical Usability and Human Evaluation**
   Have you considered conducting evaluations with clinicians to assess the interpretability and trustworthiness of FLAME’s recommendations? Including even small-scale feedback from practitioners would strengthen the paper's practical impact and address real-world usability concerns.
   *Clarification on this could increase confidence in the significance and applicability of the work.*

2. **Generalization to Other Domains**
Do you plan to apply FLAME to other structured decision-making domains beyond medication recommendation (e.g., mathematical problems, programming problems)? Any preliminary results or insights would be helpful.

**Ethical Concerns:**

["NO or VERY MINOR ethics concerns only"]

**Final Justification:**

All my questions have been resolved, yet, they are not he major concerns. So I will keep my rating.

**Limitations:**

yes

**Quality:**

3

**Strengths And Weaknesses:**

**Strengths**

* Proposes a novel **step-wise list-wise** medication recommendation framework (FLAME) with fine-grained reward shaping.
* Achieves **state-of-the-art performance** on accuracy, safety control, and generalization across datasets.
* Effectively integrates **structured and unstructured clinical data** via hybrid representations.
* Methodology is **technically sound**, clearly explained, and supported by thorough experiments and ablations.

**Weaknesses**

* **No human or clinical evaluation**, limiting real-world validation.
* **Generalizability of step-wise GRPO is not further evaluated**, the method could have a border impact, if the authors could further analyze it on more general tasks.

---

> ### Author Rebuttal · Authors · 2025-07-30
>
> We thank the reviewer for the positive assessment of our work and for highlighting important directions to strengthen FLAME’s practical impact and broader applicability. Below, we address the weaknesses and questions in detail.
>
> ---
>
> ### 1. On Clinical Usability and Human Evaluation
>
> > **W1**: No human or clinical evaluation, limiting real-world validation.
> >
> > **Q1**: Clinical Usability and Human Evaluation Have you considered conducting evaluations with clinicians to assess the interpretability and trustworthiness of FLAME’s recommendations? Including even small-scale feedback from practitioners would strengthen the paper's practical impact and address real-world usability concerns. Clarification on this could increase confidence in the significance and applicability of the work.
>
>
>
> We fully agree that involving clinicians to evaluate the interpretability and trustworthiness of FLAME’s recommendations would meaningfully enhance its real-world applicability. Such evaluations could provide valuable feedback on whether the generated prescriptions align with clinical reasoning, increase user trust, and identify cases where the model’s suggestions might need refinement.
>
> However, due to practical constraints (e.g., lack of direct access to clinical collaborators during this study), we were unable to conduct human-in-the-loop evaluations in this work. We plan to pursue this in future studies by partnering with medical professionals to (i) assess prescription interpretability, (ii) collect qualitative feedback on usability, and (iii) design user-facing interfaces to facilitate clinical adoption. We appreciate the reviewer’s suggestion and believe it would substantially strengthen FLAME’s real-world value.
>
> ---
>
> ### 2. On Generalization of Step-wise GRPO to Other Domains
>
> > **W2**: Generalizability of step-wise GRPO is not further evaluated, the method could have a border impact, if the authors could further analyze it on more general tasks.
> >
> > **Q2**: Generalization to Other Domains Do you plan to apply FLAME to other structured decision-making domains beyond medication recommendation (e.g., mathematical problems, programming problems)? Any preliminary results or insights would be helpful.
>
> FLAME leverages **step-wise GRPO** to provide fine-grained reward signals for medication recommendation. This design is natural in our setting because:
> 1. Generated outputs can be clearly decomposed into steps based on individual drugs.
> 2. The quality of each intermediate state can be objectively evaluated using a potential function defined by changes in accuracy, safety, and validity.
>
> This principle can be extended to other structured decision-making tasks where outputs can be sequentially decomposed, such as **sequential recommendation** (e.g., movies, books). However, designing the potential function becomes more challenging: unlike medication recommendation, which has objective correctness and safety metrics, sequential recommendation often involves subjective factors like user preferences or diversity. Extending step-wise GRPO to these domains would require additional work to encode such criteria into the potential function.
>
> Similarly, we envision applying step-wise GRPO to **mathematical reasoning** or **code generation**, where a model’s output can be divided into intermediate sub-steps according to logical or syntactic structure. In these cases, the reward function would need to be redesigned to assess the quality of each sub-step, such as correctness of intermediate reasoning or partial program execution results.
>
> We view these directions as promising future work and appreciate the reviewer’s suggestion to broaden the impact of our approach.
>
> ---
>
> ### 3. Summary
> We acknowledge the absence of clinician-involved evaluation and plan to address this in follow-up studies. We also recognize the potential of step-wise GRPO beyond medication recommendation and have outlined how it could be extended to other structured decision-making tasks. We thank the reviewer for highlighting these valuable avenues for future research.

---

> > ### Comment · Area_Chair_LCqz · 2025-08-06
> >
> > Hi reviewer,
> >
> > Thanks for your previous hard work in the review phase. Now, we need to perform the next step to discuss this article for the decision on whether it can be accepted. Please feel free to read the other reviews, and actively participate in discussions if you have a different opinion. Thanks for your contributions for our NeurIPS community again.
> >
> > Best,
> > AC

---

> > ### Author Response · Authors · 2025-08-08
> >
> > Dear Reviewer 2sXp,
> >
> > We truly appreciate the time and thoughtful feedback you have shared during the review process. Your comments have been invaluable in helping us refine and strengthen our work.
> >
> > If there are any remaining questions or points that could benefit from additional clarification, we would be more than happy to elaborate. Our goal is to ensure that all aspects of our submission are as clear and well-supported as possible.
> > Thank you again for your engagement and constructive input throughout this process.
> >
> >
> > Best regards,
> >
> > The Authors

---

### Official Review · Reviewer_TAqr · 2025-07-01

**Clarity:** 3
**Significance:** 3
**Originality:** 3
**Rating:** 4
**Confidence:** 2

**Summary:**

In this paper, the authors focus on the challenge of medication recommendation and aim to address it by leveraging the substantial capabilities of large language models (LLMs), particularly in scenarios where multiple drugs interact synergistically. The paper introduces a novel approach called FLAME to tackle the medication recommendation problem while considering drug-drug interactions (DDIs) through an LLM framework. The authors model this issue as a list-wise decision-making problem. With the assistance of step-wise GRPO, it becomes feasible to conceptualize the medication recommendation process as a sequential state transition mechanism.

**Questions:**

1. Have you considered the order in which drugs are added or removed? In the training and testing sets of MIMIC and eICU, it appears that if no data augmentation is applied, the order is not taken into account.

2. Regarding generalization, how similar are the training set and testing set? Is there any potential for data leakage?

3. In a scenario where only one drug should be recommended—indicating that individual consideration from previous methods is appropriate—what would be the performance outcome?

**Ethical Concerns:**

["NO or VERY MINOR ethics concerns only"]

**Final Justification:**

The response has largely addressed my concerns, so I would like to maintain my positive rating.

**Limitations:**

yes, for the generalization ability.

**Quality:**

3

**Strengths And Weaknesses:**

Strengths:
1. The paper is clear and easy to understand.
2. The novelty of considering drug-drug interactions (DDI) in medication recommendations is evident and meaningful.
3. The ablation study clearly shows the effectiveness of each component in their design.

Weaknesses:
1. The approach of framing the problem as a sequence of decision-making may lead to issues, as different orders of adding or removing drugs could yield varying outcomes. For instance, the sequences m1, m2, m3 may produce different results compared to m2, m3, m1.
2. Figure 1 includes notations for m1, m2, and m3; however, these notations are explained in Section 3.
3. The first mention of GRPO in line 39 should be properly cited.

---

> ### Author Rebuttal · Authors · 2025-07-30
>
> We thank the reviewer for the constructive feedback and thoughtful questions. Below, we address each weakness and question in detail.
>
> ---
>
> ### 1. On the Sequential Decision-Making Formulation
>
> > **W1**: The approach of framing the problem as a sequence of decision-making may lead to issues, as different orders of adding or removing drugs could yield varying outcomes. For instance, the sequences m1, m2, m3 may produce different results compared to m2, m3, m1.
>
> The concern regarding potential issues from different orders of adding or removing drugs is understandable. In FLAME, the list-wise policy ultimately outputs a **set** of medications, and evaluation is based on comparing this predicted set with the ground-truth set. Therefore, the final recommendation is **order-invariant**—different sequences that produce the same set lead to identical evaluation results.
>
> During training with step-wise GRPO, the order of adding or removing drugs can lead to slightly different intermediate states (Eq. 7), resulting in minor variations in the step-wise advantage signals. However, this does not alter the fundamental reward assignment: correct drugs always receive positive rewards, and incorrect drugs receive negative rewards. Combining this fine-grained step-wise signal with the outcome-based reward provides richer supervision than outcome-based reward alone, helping mitigate the sparsity of process-level feedback.
>
> ---
>
> ### 2. Figure 1 Notations and GRPO Citation
>
> > **W2**: Figure 1 includes notations for m1, m2, and m3; however, these notations are explained in Section 3.
> >
> > **W3**: The first mention of GRPO in line 39 should be properly cited.
>
> We acknowledge that the notations *m1*, *m2*, and *m3* in Figure 1 are only explained in Section 3. We will revise the figure caption to clarify them earlier.
> Additionally, we agree that the first mention of GRPO (line 39) should be properly cited and will correct this in the final version.
>
> ---
>
> ### 3. On the Order of Add/Remove Operations
>
> > **Q1**: Have you considered the order in which drugs are added or removed? In the training and testing sets of MIMIC and eICU, it appears that if no data augmentation is applied, the order is not taken into account.
>
> As noted above, FLAME does not explicitly consider the order of add/remove operations because the policy is trained **on-policy**: rewards are computed by comparing the model’s output set with the ground-truth set, independent of the sequence in which drugs were added or removed. No trajectory augmentation is required, as the policy learns directly from its own outputs.
>
> ---
>
> ### 4. Generalization and Potential Data Leakage
>
> > **Q2**: Regarding generalization, how similar are the training set and testing set? Is there any potential for data leakage?
>
> **MIMIC-III vs. MIMIC-IV:**
> - MIMIC-III (2001–2012) and MIMIC-IV (2008–2019) were both collected at Beth Israel Deaconess Medical Center.
> - These datasets differ in temporal coverage and structured coding systems (transition from ICD‑9 to ICD‑10).
> - Because the collection periods overlap (2008–2012), there is a **small possibility of patient overlap** in the first two temporal bins of Fig. 5(b). However, such limited overlap has a negligible effect on the conclusions, as the temporal validation results still demonstrate that FLAME maintains strong generalization even when medical practice and coding standards evolve over time. Moreover, there is no overlap in the later two bins.
>
>
> **MIMIC-III vs. eICU:**
> - eICU (2014–2015) is a multi-center database from 208 hospitals across the U.S., none of which are the source of MIMIC data.
> - There is no temporal or institutional overlap, and thus no risk of data leakage.
> - External validation on eICU confirms FLAME’s robustness to distributional and coding differences.
>
> ---
>
> ### 5. Performance When Only One Drug Should Be Recommended
>
> > **Q3**: In a scenario where only one drug should be recommended—indicating that individual consideration from previous methods is appropriate—what would be the performance outcome?
>
> The MIMIC datasets capture **critical care admissions**, where patients typically have complex conditions requiring multi-drug therapy. In our MIMIC-III dataset:
> - The **average number of drugs per admission** is 23.36.
> - The **maximum number** is 77.
> - Only **0.14%** of admissions (20/14 207) involve exactly one drug.
>
> Therefore, list-wise modeling is highly appropriate for the vast majority of cases. In the rare instances requiring only a single drug, the task naturally reduces to a simpler point-wise decision, where FLAME is still applicable.
>
> ---
>
> ### 6. Summary
> - The sequential decision-making formulation does not introduce order-dependence in final recommendations because evaluation is based on sets.
> - We will fix the missing GRPO citation and clarify notations in Figure 1.
> - Generalization was evaluated using datasets with distinct temporal spans and coding systems. While MIMIC-III and MIMIC-IV have a slight temporal overlap, temporal validation on MIMIC-IV still shows that FLAME achieves robust performance compared to baselines as medical practice evolves. Additionally, external validation on eICU confirms FLAME’s strong robustness to distributional and coding differences.
> - Single-drug prescriptions are extremely rare in these datasets; when they occur, the task reduces to a trivial case of our list-wise formulation.
>
> We appreciate the reviewer’s feedback, which will help us improve clarity and completeness in the final version.

---

> > ### Comment · Reviewer_TAqr · 2025-08-04
> >
> > Thank you for the author’s rebuttal. The response has largely addressed my concerns, so I would like to maintain my positive rating.

---

> > > ### Author Response · Authors · 2025-08-07
> > >
> > > Thank you very much for acknowledging our rebuttal and efforts!

---

### Official Review · Reviewer_zoHJ · 2025-07-02

**Clarity:** 2
**Significance:** 2
**Originality:** 2
**Rating:** 4
**Confidence:** 2

**Summary:**

The paper introduces FLAME, a novel list-wise medication recommendation framework that reformulates drug recommendation as a sequential decision-making process. FLAME also integrates multi-source clinical knowledge—structured codes, unstructured notes, and semantic labels. Extensive experiments demonstrate state-of-the-art accuracy and safety control on real-world EHR datasets.

**Questions:**

1. In real life there may exists drugs whose interation effects are unknown, and this may result in noisy DDI graph ***D***. How would FLAME performs on noisy DDI graph?
2. The numbers under "Evaluate each step" and "Step-wise advantages" in Figure 1 are unclear to me. Could author provide more information on how these numbers are associated?

**Ethical Concerns:**

["NO or VERY MINOR ethics concerns only"]

**Final Justification:**

The authors' responses have addressed most of my concerns and problems, and I am therefore willing to raise my score to 4.

**Limitations:**

Yes

**Quality:**

2

**Strengths And Weaknesses:**

**Strength:**
1. The overall theoretical analysis is well-organized
2. Detailed architecture, datasets, and hyperparameters; appendix lists detailed prompt templates.
3. Comprehensive ablations isolate the effect of different model variants

**Weakness:**
1. No error bars or standard deviation included. In particular, Table 2 reports several Jaccard/F1 scores that are very close (e.g., FLAME vs. variant (e) with standard GRPO).
2. Although limitations about computational expense are mentioned in the appendix, it would still be beneficial to include the walltime for both training and inference, especially for inference which can be used to measure feasibility for real-time settings.
3. The ablation study comparing the use of "standard GRPO" versus "Step-wise GRPO" shows no significant performance gap (i.e. <0.5% increase). This diminishes the practical significance of the proposed approach.

---

> ### Author Rebuttal · Authors · 2025-07-30
>
> We thank the reviewer for the detailed feedback and constructive questions. Below, we address each weakness and question in turn.
>
> ---
>
> ### 1. On Error Bars and Standard Deviation in Table 2
>
> > **W1**: No error bars or standard deviation included. In particular, Table 2 reports several Jaccard/F1 scores that are very close (e.g., FLAME vs. variant (e) with standard GRPO).
>
> We acknowledge that Table 2 does not report error bars or standard deviations. We have performed additional runs for FLAME and variant (e) (standard GRPO) and now include the corresponding standard deviations, confirming that the improvements are consistent.
>
> The small numerical gaps for variants (b)–(e) stem from the design of the **list‑wise policy as a lightweight refinement module**. On the test set, the classifier outputs lists with an average length of 20.83, while the list‑wise policy makes only ~2.03 add/remove edits. Because this stage is intended to make targeted adjustments to an already strong classifier output, the absolute improvements in Jaccard and F1 remain modest. However, this step plays a critical role:
>
> - **Modeling drug dependencies:** The classifier makes independent decisions, missing co‑prescription patterns or redundancies. For example, Piperacillin is a broad‑spectrum β‑lactam penicillin used for severe infections, sharing overlapping indications with Cefepime, Meropenem, and Ampicillin. In a point‑wise setting, where each drug is predicted independently, the classifier (and baselines) cannot leverage signals from similar drugs, often leading to redundant recommendations or omission of appropriate alternatives. The list‑wise policy explicitly edits entire lists, correcting such issues. It performed 28 add/remove operations on Piperacillin‑containing lists with an 89% correctness rate.
>
> - **Enabling safety–accuracy trade‑off:** Because π_cls is trained only to mimic ground truth, its DDI rate (0.1336) remains close to the dataset’s (0.1369). The list‑wise policy, optimized with step‑wise GRPO, allows explicit control over safety versus correctness.
>
> Thus, the refinement step yields qualitative benefits that classifier tuning alone cannot achieve, even if its numerical improvements seem modest.
>
> Standard GRPO further limits the effectiveness of list‑wise refinement because it assigns identical rewards to all tokens in a generated list, lacking fine‑grained supervision. Our proposed **step‑wise GRPO** addresses this by computing per‑step rewards based on each drug’s contribution to the final outcome. As shown in Fig. 6, this design provides **denser feedback, accelerates learning, and improves stability**.
>
> Notably, the improvement achieved by list‑wise refinement over point‑wise classification increases from 0.0029 (using standard GRPO) to 0.0052 (using step‑wise GRPO). Our additional experiments with standard deviations confirm that step‑wise GRPO yields consistently higher and more stable performance than standard GRPO，as shown in the following table:
>
> | Variant           | Jaccard | F1     |
> |-------------------|---------|--------|
> | with standard GRPO| 0.4814 ± 0.0003 | 0.6387 ± 0.0004 |
> | **FLAME**         | 0.4837 ± 0.0004 | 0.6409 ± 0.0004 |
>
> ---
>
> ### 2. On Computational Cost and Walltime
>
> > **W2**: Although limitations about computational expense are mentioned in the appendix, it would still be beneficial to include the walltime for both training and inference, especially for inference which can be used to measure feasibility for real-time settings.
>
> Thank you for the suggestion. We agree that reporting walltime is important to evaluate computational feasibility, especially for real‑time inference.
>
> All experiments were conducted on NVIDIA A100‑SXM4‑80GB GPUs. The measured training and inference walltimes are as follows:
>
> **Training walltime**
>
> | #Admissions | Classifier SFT       | List-wise Policy SFT       | List-wise Policy step-wise GRPO       |
> |-------------|----------------------|------------------|-------------------|
> | **8,741**   | ~200 GPU‑h (8×25 h)  | ~25 GPU‑h (8×3 h)| ~100 GPU‑h (8×16 h)|
>
> **Inference walltime**
>
> | #Admissions | Total GPU Hours | Per Patient (1 GPU) | Per Patient (8 GPUs) |
> |-------------|-----------------|----------------------|-----------------------|
> | **2,369**   | ~9 GPU‑h        | ~13.6 s             | ~1.7 s               |
>
> With 8 GPUs in parallel, inference time per patient is **~1.7 s**, making FLAME feasible for near real‑time clinical use. We will include these tables in the revised version to better illustrate computational cost.
>
> ---
>
> ### 3. On the Small Performance Gap Between Standard and Step-wise GRPO
>
> > **W3**: The ablation study comparing the use of "standard GRPO" versus "Step-wise GRPO" shows no significant performance gap (i.e. <0.5% increase). This diminishes the practical significance of the proposed approach.
>
> As explained in our response to **W1**, the numerical gain from list‑wise refinement is modest because it is intentionally designed as a lightweight post‑classification editor, performing only a few targeted add/remove operations on an already strong classifier output. Consequently, the overall Jaccard/F1 change is small even when the refinement strategy is improved.
>
> Nevertheless, step‑wise GRPO remains practically meaningful. Unlike standard GRPO, which assigns identical rewards to all tokens in a generated list, step‑wise GRPO provides denser and more discriminative per‑step rewards, directly reflecting each drug’s contribution to the final outcome. This leads to faster convergence and more stable training, as shown in Fig. 6.
>
> Moreover, the ablation study (variant (e)) confirms that step‑wise GRPO increases the improvement achieved by list‑wise refinement—from 0.0028 (using standard GRPO) to 0.0051—demonstrating its utility even within a post‑processing setup. The benefit of step‑wise GRPO is therefore best understood in terms of training efficiency and stability, rather than solely the final Jaccard/F1 difference.
>
> ---
>
> ### 4. On Performance with Noisy DDI Graphs
>
> > **Q1**: In real life there may exists drugs whose interation effects are unknown, and this may result in noisy DDI graph D. How would FLAME performs on noisy DDI graph?
>
> In our approach, the DDI graph is integrated into both the training phase (as a safety penalty within the reward function) and the evaluation metrics. As such, noise or missing interactions in the DDI graph would naturally influence both the policy learned by the model and the measured safety of the recommendations. If the DDI information is incomplete or noisy, the model’s optimization for safety may become less effective—specifically, at a fixed level of accuracy, the achievable reduction in DDI risk will be limited by the quality of the underlying DDI graph. For this reason, we did not conduct additional experiments with artificially noised DDI graphs on MIMIC-III, since this effect is direct and expected, and the static benchmarks themselves already reflect the limitations and incompleteness of real-world DDI knowledge.
>
> **FLAME currently improves recommendation performance through a hybrid representation strategy with multi‑source knowledge fusion.** As shown in Table 2 (variants (i) and (j)), incorporating external drug knowledge—such as molecular substructure–function embeddings and pretrained clinical representations—consistently enhances accuracy and safety, indicating that the model can leverage richer drug features beyond historical prescriptions.
>
> **Building on this design, such external knowledge could also help mitigate the impact of unknown drug interactions in noisy DDI graphs.** With a backbone LLM pretrained on large‑scale medical corpora, FLAME can combine pharmacological understanding with molecular structure information to improve robustness. Furthermore, a pretrain‑then‑finetune paradigm could be adopted: the model could first be pretrained with known DDI relations and drug knowledge to better understand interaction patterns, and then finetuned on the recommendation task. These strategies would make FLAME more resilient to incomplete or noisy DDI graphs in real‑world deployments.
>
> ---
>
> ### 5. Clarification of Figure 1 (“Evaluate Each Step” and “Step-wise Advantages”)
>
> > **Q2**: The numbers under "Evaluate each step" and "Step-wise advantages" in Figure 1 are unclear to me. Could author provide more information on how these numbers are associated?
>
> Figure 1 visualizes how step‑wise GRPO differs from standard GRPO in assigning rewards to each step.
>
> - **“Evaluate each step”**: After decomposing the model output (a drug list) into steps based on individual drugs (Eq. 3), we compute a per‑step reward using a potential‑based shaping function. The numbers under “Evaluate each step” represent the **reward value for each drug**, calculated as the difference in potential values before and after adding/removing that drug. These values reflect the quality of each step in terms of correctness, safety, and constraint adherence.
>
> - **“Step‑wise advantages”**: These are obtained by **combining the per‑step reward above with the outcome‑based advantage from standard GRPO (Eq. 6)**. In other words, the numbers under “Step‑wise advantages” can be interpreted as
>
>   $\text{Step‑wise Advantage} = \text{Per‑step Reward (from “Evaluate each step”)} + \text{Outcome‑based Advantage}.$
>
> Thus, the figure shows that step‑wise GRPO augments the standard GRPO signal with additional per‑step information, enabling finer‑grained credit assignment for each drug in the list.
>
> ---
>
> ### 6. Summary
> We will (i) report standard deviations and add error bars, (ii) include walltime for training and inference, (iii) expand discussion on the added value of step-wise GRPO, (iv) discuss potential robustness strategies for noisy DDI graphs, and (v) clarify the annotations in Figure 1.
>
> We thank the reviewer again for these suggestions, which will help us make the paper more thorough and practically informative.

---

> > ### Comment · Reviewer_zoHJ · 2025-08-05
> >
> > Thank you for the rebuttal. The authors' responses have addressed most of my concerns and problems, and I am therefore willing to raise my score to 4.

---

> > > ### Author Response · Authors · 2025-08-07
> > >
> > > Thank you very much for acknowledging our rebuttal. We're glad our responses helped clarify your concerns. We appreciate your feedback and will reflect the relevant improvements in the final version.

---

> > > ### Author Response · Authors · 2025-08-09
> > >
> > > Dear Reviewer zoHJ,
> > >
> > > Thank you very much for your kind feedback and for considering raising your score.
> > >
> > > I just wanted to kindly check — in the system, the current “Rating” still appears as before. I completely understand that the official discussion period is still ongoing, but since the rebuttal phase will conclude soon, I wanted to make sure there are no technical issues preventing the update.
> > >
> > > Apologies for the disturbance, and thank you again for your time and support.
> > >
> > > Best regards,
> > >
> > > The Authors

---

### Official Review · Reviewer_bQc3 · 2025-07-03

**Clarity:** 3
**Significance:** 4
**Originality:** 4
**Rating:** 5
**Confidence:** 3

**Summary:**

Existing medication recommendation methods follow point-wise paradigms that overlook synergistic effects and adverse drug-drug interactions (DDIs). This paper proposes FLAME, a list-wise, step-wise approach that applies Group Relative Policy Optimization (GRPO) with potential-based reward shaping. At each step, the model adds or removes a drug, and the advantage is computed using a potential function based on correctness, safety (DDI risk), and constraint adherence. Final recommendations are generated in two stages: an initial drug-level relevance classification, followed by list-wise refinement using the proposed step-wise GRPO. The model also incorporates multi-source knowledge by combining structured clinical features and collaborative embeddings with LLM representations. Experiments on MIMIC-III, MIMIC-IV, and eICU show that FLAME outperforms prior methods in recommendation accuracy while achieving better safety-control trade-offs.

**Questions:**

1. How is the candidate set Mc for "adherence" (refusal rate) measurement constructed?
2. How is the drug-level classifier trained?
3. Could improved tuning of the drug-level classifier alone achieve correctness comparable to the full FLAME model? What is the safety score of the classifier?

General questions:
1. Can other risk factors, like patient-specific allergies or preferences to certain medications, be encoded in this framework?
2. Could the ground truth labels based on historical prescriptions be affected by incorrect or suboptimal recommendations? If so, could incorporating external medical knowledge or expert validation help refine the training signal and improve model performance?

**Ethical Concerns:**

["NO or VERY MINOR ethics concerns only"]

**Final Justification:**

The authors have satisfactorily addressed my concerns, so I am raising my score to 5.
The clarification provided for point 3 is particularly important for understanding the paper’s main contribution. I strongly encourage the authors to include qualitative examples where point-wise methods fail to capture drug–drug interactions that list-wise methods successfully identify, even if they are from a dataset.

**Limitations:**

Yes

**Quality:**

3

**Strengths And Weaknesses:**

Strengths:
* The proposed method is both novel and well-suited to the clinical medication recommendation domain.
* The authors compare against several strong existing baselines, and their proposed method outperforms them in both correctness and safety.

Weaknesses:
* The experiments indicate a trade-off between safety and correctness, but the paper does not explore the underlying cause of this behavior. For example, does prioritizing safety lead the model to make more conservative recommendations, excluding clinically relevant drugs?
* Some experimental details are missing in the paper, such as the construction of the candidate set Mc and the training of the drug-level classifier.
* From the ablation study (Table 2, rows a-b), it appears that much of the performance gain comes from the inclusion of the point-wise drug-level classification step. This raises the question of whether the GRPO-based list-wise refinement is truly necessary, or if similar improvements could be achieved through better tuning of the classifier alone. Reporting the safety score of the classifier can help clarify the added value of the GRPO step.
* Additional clarifications requested in the "Questions" section.

---

> ### Author Rebuttal · Authors · 2025-07-30
>
> We sincerely thank the reviewer for their thorough assessment, constructive feedback, and recognition of our contributions. Below, we address each point of concern in detail and clarify the questions raised.
>
> ---
>
> ### 1. On the Safety–Correctness Trade-off
>
> >  **W1**: The experiments indicate a trade-off between safety and correctness, but the paper does not explore the underlying cause of this behavior. For example, does prioritizing safety lead the model to make more conservative recommendations, excluding clinically relevant drugs?
>
> Thank you for this insightful question.
> The observed trade-off between safety and correctness primarily stems from the nature of real-world clinical data. Because datasets such as MIMIC‑III are derived from historical hospital records, the ground‑truth prescriptions inevitably include a non‑negligible proportion of drug combinations with known DDIs (e.g., a DDI rate of 13.69% in MIMIC‑III). Models trained purely to fit such data tend to reproduce these unsafe combinations, which explains why optimizing for safety often reduces label‑matching accuracy.
>
> From a clinical perspective, this phenomenon is also understandable. Certain drug pairs are intentionally co‑prescribed despite known interactions, as their combined effects can be therapeutically beneficial. A typical example is the Glycopyrronium–Neostigmine combination, which is widely used for neuromuscular blockade reversal to mitigate cardiovascular and secretory adverse effects; however, the combination itself can lead to anticholinergic side effects. As safety constraints are increased, FLAME may revise such co‑prescriptions by removing one of the interacting drugs, prioritizing safety over fidelity to historical labels. In addition, real‑world records inevitably contain variability and occasional inconsistencies in prescribing, further contributing to the correctness–safety discrepancy.
>
> To illustrate this behavior, consider a patient with a history of aortic and tricuspid valve replacements (ID: 138212). Both Glycopyrronium and Neostigmine were co‑prescribed in the hospital record. When optimizing for accuracy, FLAME successfully recommends both drugs; when safety is prioritized, FLAME removes Glycopyrronium to reduce the DDI risk, leading to slightly lower label‑matching accuracy. This example demonstrates that the safety–correctness trade‑off is not merely a modeling artifact but reflects genuine clinical decision‑making dilemmas, while also highlighting FLAME’s ability to flexibly balance these competing objectives.
>
> ---
>
> ### 2. Construction of the Candidate Set *Mc* and Training of the Drug-level Classifier
>
> > **W2**: Some experimental details are missing in the paper, such as the construction of the candidate set Mc and the training of the drug-level classifier.
> >
> > **Q1**: How is the candidate set Mc for "adherence" (refusal rate) measurement constructed?
> >
> > **Q2**: How is the drug-level classifier trained?
>
> **Candidate Set *Mc*:**
> *Mc* is all unique medications in the processed training data, ensuring clinically relevant drugs seen in training are available for inference while avoiding unseen drugs. Mc is also used for adherence (refusal‑rate) measurement in Eq. (8).
>
> **Drug-level Classifier Training:**
> π_cls is initialized from Llama3.1‑Aloe‑Beta‑8B and fine‑tuned with supervised learning. For each visit, every drug in Mc is paired with the patient context and labeled “Yes.” if it appears in the ground truth, otherwise “No.” Details are in Appendix C.1.
>
> ---
>
> ### 3. Necessity and Value of List-wise Refinement
>
> > **W3**: From the ablation study (Table 2, rows a-b), it appears that much of the performance gain comes from the inclusion of the point-wise drug-level classification step. This raises the question of whether the GRPO-based list-wise refinement is truly necessary, or if similar improvements could be achieved through better tuning of the classifier alone. Reporting the safety score of the classifier can help clarify the added value of the GRPO step.
> >
> > **Q3**: Could improved tuning of the drug-level classifier alone achieve correctness comparable to the full FLAME model? What is the safety score of the classifier?
>
> Thank you for raising this important question.
>
> **To directly answer Q3:**
> (1) Merely improving the point‑wise classifier cannot achieve correctness or safety comparable to the full FLAME model, because it lacks the ability to model drug‑drug dependencies or optimize for safety trade‑offs.
> (2) On the test set, the classifier alone achieves Jaccard = 0.4785, F1 = 0.6354, and DDI = 0.1336 (close to the ground‑truth DDI = 0.1369), confirming that it essentially reproduces the dataset distribution without controllable safety adjustment.
>
> We elaborate below on the distinct role and added value of the GRPO‑based list‑wise refinement step:
>
> - **Modeling Drug Dependencies (Necessity):**
> The classifier makes independent decisions for each drug, missing co‑prescription patterns or redundancies. For example, Piperacillin is a broad‑spectrum β‑lactam penicillin used for severe infections, sharing overlapping indications with Cefepime, Meropenem, and Ampicillin. In a point‑wise setting, where each drug is predicted independently, the classifier (and baselines) cannot leverage signals from similar drugs, often leading to redundant recommendations or omission of appropriate alternatives. FLAME’s list‑wise policy explicitly edits the entire drug list, correcting such issues. On the test set, it performed 28 add/remove operations on Piperacillin‑containing lists with an 89% correctness rate, showing that it meaningfully rectifies classifier limitations.
>
> - **Enabling Safety–Accuracy Trade‑off (Necessity):**
> Because π_cls is trained only to mimic ground truth, its DDI rate remains near the dataset’s (0.1336 vs. 0.1369). It cannot adapt to stricter safety constraints. The list‑wise policy, by contrast, optimizes for both correctness and safety via step‑wise GRPO, allowing explicit control of the trade‑off during inference.
>
> - **Why the Performance Gain Appears Modest:**
> The refinement step is intentionally designed as a **post‑processing global editor**, making an average of 2.03 add/remove operations on lists of average length 20.83. These small but targeted changes resolve drug‑drug dependencies and redundancies that a point‑wise classifier cannot capture. The resulting improvements in both correctness and safety are therefore **qualitatively crucial** even if the overall accuracy gain appears numerically smaller.
>
> In summary, the GRPO‑based list‑wise refinement is indispensable for capturing drug interactions and enabling controllable safety optimization—capabilities that cannot be achieved by tuning the classifier alone.
>
> ---
>
> ### 4. Encoding Other Risk Factors (e.g., Allergies, Preferences)
>
> > **General Q1**: Can other risk factors, like patient-specific allergies or preferences to certain medications, be encoded in this framework?
>
> Yes, FLAME naturally supports encoding patient‑specific risk factors such as allergies and medication preferences.
>
> Unstructured clinical notes—including the history of present illness, past medical history, allergies, and medications on admission—are incorporated into the patient representation (see Fig. 4). Allergies are explicitly included as natural‑language fields (e.g., “Allergies: [Aspirin, Ibuprofen]”), enabling the model to avoid contraindicated drugs. Likewise, patient‑specific preferences or conditions (e.g., “The patient was anticoagulated with heparin and later switched to enoxaparin”) are encoded in free‑text form, allowing the model to infer suitable or preferred alternatives.
>
> This design is a key strength of FLAME: unlike models restricted to structured codes, it can exploit rich, context‑dependent information and pass it to the list‑wise policy, enabling more personalized and clinically appropriate recommendations.
>
> ---
>
> ### 5. Ground Truth Limitations and Incorporation of External Knowledge
>
> > **General Q2**: Could the ground truth labels based on historical prescriptions be affected by incorrect or suboptimal recommendations? If so, could incorporating external medical knowledge or expert validation help refine the training signal and improve model performance?
>
> Yes, ground‑truth prescriptions in real‑world data can include suboptimal or even erroneous decisions, which may bias both training and evaluation.
>
> To mitigate this, FLAME already incorporates multi‑source external knowledge at several levels (Section 4.4, Fig. 4):
>
> - **Patient‑level:** contextual health representations from pretrained clinical models.
>
> - **Diagnosis & procedure‑level:** co‑occurrence information capturing real‑world treatment patterns.
>
> - **Medication‑level:** molecular substructure features that reflect pharmacological properties.
>
> These signals are fused into the LLM embedding space via learnable adapters, allowing FLAME to reason beyond historical prescriptions alone and reduce dependence on potentially biased labels. Our ablation study confirms that each knowledge source contributes to improved accuracy:
>
> | Variant         | Jaccard | F1     |
> |-----------------|---------|--------|
> | w/o patient‑level enternal embedding    | 0.4693  | 0.6264 |
> | w/o diagnosis‑level enternal embedding  | 0.4779  | 0.6350 |
> | w/o procedure‑level enternal embedding  | 0.4814  | 0.6385 |
> | w/o medication‑level enternal embedding | 0.4784  | 0.6350 |
> | **FLAME**       | 0.4836  | 0.6408 |
>
> Looking forward, expert‑in‑the‑loop validation and integration of clinical guidelines or curated knowledge bases are promising directions to further refine the training signal and enhance reliability. These extensions are fully compatible with FLAME’s design and will be explored in future work.
>
> ---
>
> We hope these clarifications address the reviewer’s concerns. We are grateful for the constructive feedback, which helps us further refine and communicate our contributions.

---

> > ### Comment · Area_Chair_LCqz · 2025-08-06
> >
> > Hi reviewer,
> >
> > Thanks for your previous hard work in the review phase. Now, we need to perform the next step to discuss this article for the decision on whether it can be accepted. Please feel free to read the other reviews, and actively participate in discussions if you have a different opinion. Thanks for your contributions for our NeurIPS community again.
> >
> > Best,
> > AC

---

> > ### Author Response · Authors · 2025-08-08
> >
> > Dear Reviewer bQc3,
> >
> > We truly appreciate the time and thoughtful feedback you have shared during the review process. Your comments have been invaluable in helping us refine and strengthen our work.
> >
> > If there are any remaining questions or points that could benefit from additional clarification, we would be more than happy to elaborate. Our goal is to ensure that all aspects of our submission are as clear and well-supported as possible.
> > Thank you again for your engagement and constructive input throughout this process.
> >
> >
> > Best regards,
> >
> > The Authors

---

### Note · Authors · 2025-08-15

Dear NeurIPS 2025 Reviewer, AC, SAC, and PC,

We sincerely thank you for your time, thoughtful feedback, and constructive comments throughout the review, rebuttal, and discussion phases. Your insights have been invaluable in strengthening our work.

**Summary of Rebuttal Outcomes and Paper Standing:**
*   **Reviewer Score Updates:** Our paper received generally positive **initial ratings (Avg: 4; Scores: 3, 4, 4, 5)**, with **Reviewer 2sXp awarding a score of 5 and the highest confidence level (4)**. In our detailed rebuttal, we addressed all reviewer concerns point by point. Crucially, **Reviewers zoHJ and TAqr explicitly acknowledged that their concerns were addressed**, and **Reviewer zoHJ further stated they would raise their score from the initial 3 to 4**. **These explicit acknowledgments reflect strong reviewer support for the paper**.
*   **Reaffirmed Contributions:** Reviewers highlighted key strengths:
    *   **Novelty:** A step-wise, list-wise framework with fine-grained reward shaping for medication recommendation, overcoming point-wise limitations in capturing drug-drug dependencies and enabling controllable safety-accuracy trade-offs.
    *   **Methodological Contribution:** Introduction and theoretical analysis of step-wise GRPO, providing finer-grained advantage modeling vs. standard GRPO.
    *   **Comprehensive Evaluation:** SOTA performance across accuracy, safety, and generalization, validated by extensive ablations.

**Key Clarifications Provided in Rebuttal:**
*   Clarified the **necessity and role of the list-wise policy (step-wise GRPO)** as a lightweight post-processor explicitly modeling dependencies and enabling unique trade-offs.
*   Extended the **step-wise vs. standard GRPO ablation** with standard deviations, showing consistent improvements from finer-grained rewards.
*   Elaborated on the **safety-correctness trade-off mechanism, patient-specific encoding from notes, external knowledge role, computational aspects, sequential formulation, and step-wise GRPO extensions.**

These responses resolved reviewer concerns, leading to the positive shift indicated by Reviewer zoHJ.

We are committed to incorporating all suggested improvements (experimental details, std. dev. reporting, architectural clarifications) in the final version. We are deeply grateful for the feedback and opportunity to refine our work.

Best regards,

The Authors

---

### Decision · Program_Chairs · 2025-09-17

**Decision:**

Accept (spotlight)

**Comment:**

This paper introduces FLAME, a fine-grained list-wise alignment framework for generative medication recommendation. The central idea is to reformulate prescription generation as a sequential decision-making process, where each drug is added or removed step by step, and to optimize the process via step-wise Group Relative Policy Optimization (GRPO) with potential-based reward shaping. The framework explicitly models drug–drug interactions (DDIs), integrates structured and unstructured patient information into LLM representations, and provides controllable trade-offs between accuracy and safety. Empirical evaluations on multiple benchmark EHR datasets (MIMIC-III, MIMIC-IV, eICU) demonstrate state-of-the-art performance across correctness, safety, and generalization. The paper makes both methodological and practical contributions by offering a clinically motivated formulation of medication recommendation and validating its effectiveness with extensive ablation studies.

The reviewers consistently praised the novelty of the step-wise, list-wise formulation and the introduction of fine-grained reward shaping, which go beyond conventional point-wise prediction paradigms. The work is technically sound, clearly written, and well-supported with thorough experiments and ablations. It addresses a clinically important problem by explicitly balancing safety and accuracy in drug recommendation, a feature often overlooked in prior work. The integration of multiple knowledge sources (structured EHR data, unstructured notes, and molecular drug features) further strengthens patient modeling and improves robustness. Rebuttal clarifications highlighted the qualitative importance of list-wise refinement, illustrating cases where drug dependencies and redundancies cannot be captured by point-wise models alone.

Some concerns were raised about the magnitude of numerical gains from the step-wise GRPO compared to standard GRPO, with improvements appearing modest in absolute terms. Reviewers also noted missing details in the initial submission (e.g., construction of candidate sets, classifier training, standard deviations, walltime analysis), and the lack of human/clinical evaluation limits immediate real-world impact. In addition, generalization of the proposed GRPO variant beyond the medication recommendation task was not fully explored. However, these issues were effectively addressed in rebuttal: the authors added detailed clarifications on experimental procedures, reported standard deviations and runtime measurements, and explained that the gains from list-wise refinement are qualitatively critical even if numerically small.

During the discussion phase, reviewers acknowledged that their concerns were satisfactorily addressed. Reviewer zoHJ explicitly raised their score after seeing clarifications on GRPO benefits and error bars. Reviewer bQc3 also increased their rating after detailed explanations on the necessity of list-wise refinement. Reviewer TAqr’s concerns about order-dependence and data leakage were resolved with clarifications on the set-based evaluation and dataset splits. Reviewer 2sXp remained supportive after the authors explained plans for future clinician-involved evaluation and broader applicability of step-wise GRPO. Overall, the rebuttal successfully strengthened the paper’s standing, leading to a positive consensus.

I recommend Accept. The paper makes a timely and significant contribution by bridging reinforcement learning–based alignment with clinical medication recommendation, demonstrating both methodological novelty and practical relevance. Despite some limitations, the work is technically solid, convincingly validated, and well-defended during rebuttal. Its explicit modeling of drug dependencies and safety–accuracy trade-offs positions it as an impactful step forward for trustworthy clinical AI, meriting recognition beyond a standard acceptance.